# Variation of Physiological and Behavioural Parameters during the Oestrous Cycle in Mares

**DOI:** 10.3390/ani13020211

**Published:** 2023-01-06

**Authors:** Anna Stachurska, Witold Kędzierski, Beata Kaczmarek, Anna Wiśniewska, Beata Żylińska, Iwona Janczarek

**Affiliations:** 1Department of Horse Breeding and Use, Faculty of Animal Sciences and Bioeconomy, University of Life Sciences in Lublin, 20-950 Lublin, Poland; 2Department of Biochemistry, Faculty of Veterinary Medicine, University of Life Sciences in Lublin, 20-950 Lublin, Poland; 3Department and Clinic of Animal Internal Diseases, Faculty of Veterinary Medicine, University of Life Sciences in Lublin, ul. Głęboka 30, 20-612 Lublin, Poland; 4Department and Clinic of Animal Surgery, Faculty of Veterinary Medicine, University of Life Sciences in Lublin, ul. Głęboka 30, 20-612 Lublin, Poland

**Keywords:** horse, oestrus cycle, physiology, behaviour, welfare, oestrus, dioestrus

## Abstract

**Simple Summary:**

Apart from the sexual behaviour of mares in heat (oestrus), the rhythmicity of many physiological and behavioural traits during the whole oestrous cycle is scarcely known. We aimed at determining changes in cardiac activity variables, rectal and skin surface temperatures, relationship towards humans and other horses, and the time of standing and locomotor activity. Fifteen adult mares were examined in the morning and evening in oestrus (six days) and in dioestrus (five days—once every third day). Heart rate and rectal and surface temperatures are higher in the evening than in the morning and higher in oestrus than dioestrus. During oestrus, the mares show a better behaviour towards humans and other horses and are less locomotively active than during dioestrus. A rapid decrease in most of the physiological parameters occurs at the end of oestrus, whereas alterations in the behaviour towards humans and other horses appear at the beginning of dioestrus. The time of locomotor activity arises at the end of oestrus. The outcomes may contribute to the knowledge of, among others, mare owners who evaluate the oestrus by sexual behaviours without regarding other rhythmically changing variables.

**Abstract:**

The behaviour of oestrous mares is well-known in horse breeding. However, alterations in their physiological and behavioural indices during the whole oestrous cycle are scarcely known. The objective of the study was to analyse changes in cardiac activity variables, rectal and superficial temperatures, behaviour towards humans and conspecifics, and the time of standing and locomotor activity in mares during their oestrous cycle. Fifteen adult mares in oestrus were examined in the morning and evening (six successive days) and in dioestrus (five days—once every third day). The oscillation of physiological and behavioural variables accompanies changes in mares’ sexual behaviour. Most physiological variables studied in oestrus indicate the elevated activity of the adrenergic nervous system and, opposite to that, both behaviour towards humans and conspecifics and the time of standing relate to a relaxed state. The end of oestrus, manifested by a rapid decrease in most of the physiological variables studied, is followed by changes of behavioural variables at the beginning of dioestrus. The time of locomotor activity arises at the end of oestrus. The outcomes may contribute to the knowledge of, among others, mare owners who evaluate the oestrus by mares’ sexual behaviours without regarding other rhythmically changing variables.

## 1. Introduction

Behavioural differences in the trainability and personality between horses of different sexes are frequently quantified utilising questionnaires. A survey study by Fenner et al. [1], for example, found that riders prefer geldings for their calmness, trainability, reliability, and predictability compared to mares and stallions. Mares are not preferred for three equestrian disciplines: dressage, showjumping, and trail-riding. Due to their sexual behaviour, stallions are allocated to adult male riders, who are scarce concerning the predominance of women in equitation. Stallions are commonly regarded as useless for leisure riding because of their behaviour and for the sake of human safety. This opinion results in a lack of published research on adult stallion personality traits compared to mares and geldings [2]. Mares are rated as tenser, more aggressive and excitable, are found to panic more easily, and exhibit more anxiety and less affability compared to geldings [3]. However, according to a survey study of 110 estimated behavioural traits by Aune et al. [4], mares and geldings were assessed similarly for their behaviour when ridden and only three non-ridden traits were evaluated differently for these genders.

Why is the behaviour of mares often found to be more difficult when handling and riding than that of geldings? The issue is assumed to be associated with hormone levels during the oestrous cycle. Oestrus in the mare is a direct consequence of oestradiol secreted by the follicle and suppressed by the progesterone secreted by the corpus luteum. As has been learned from observing free-running horses, signs of oestrous behaviour displayed during an ovulatory period throughout the year may facilitate the maintenance of the dominance hierarchy by mares in a herd [5]. Oestrous females typically show increased frequencies of approaching and following the stallion, urinating, standing still with the hind limbs spread apart, clitoral winking, and deviating the tail from the perineum [5,6]. A free-running oestrous female is active in the stimulation of the male. Most precopulatory interactions in nature are initiated by the mare approaching the stallion as opposed to the stallion approaching the mare [7]. Dioestrus is characterised by the avoidance of an approaching stallion and aggression towards the stallion, exhibited by squealing, striking, and kicking [5]. All these facts show that the sexual behaviour in mares may cause them to be difficult to handle and have detrimental effects on performance [8]. Hence, many methods affecting horse physiology were created to suppress the undesirable behaviour in the mare [9,10,11].

Physiological and behavioural changes that occur during the oestrous cycle may be characterised by several variables, such as cardiac activity parameters, body temperature, behaviour, and locomotor activity. The cardiac activity reflects the emotional excitability, for example, differences between fillies and colts [12,13]. Cardiac variables in response to stressors in three-year-old sport horses showed mares to be more sensitive than stallions in some situations [14]. Another tool for assessing physiological changes in horses is the measurement of rectal and superficial temperatures [15,16,17]. A horse’s behavioural changes in relation to humans or conspecifics can be determined by various tests and the frequency of affiliative or agonistic behaviours [18,19,20]. The changes in behaviour may be connected with locomotor activity [21,22].

The sexual behaviour of oestrous mares is well-known in horse breeding. Changes in the mare’s physiological indices and behavioural alterations in the human–horse relationship, relationships to conspecifics, and locomotor activity during the whole oestrous cycle may provide useful information to improve animal welfare and human safety. The objective of this study was to analyse some of the physiological and behavioural changes in mares during the oestrous cycle.

## 2. Materials and Methods

### 2.1. Horses and Management

Fifteen warmblood mares, from seven to ten years old and prepared to be mated during the following oestrus, were examined. Each mare had foaled at least once during the three years before the study but none of them foaled in the spring when the investigation was conducted. The length of the cycle and heat was determined in the mares during the earlier breeding seasons. Only mares with about a 21-day oestrous cycle, including a 5–6-day heat during the breeding season, were studied; mares of shorter or longer cycles and heats were excluded from the study. The signs of oestrus were identified during daily observations by a breeder (I.J., author) in the stable and for a maximum of 10 min at a teasing board. The signs of heat considered were urinating, standing still with the hind limbs spread apart, clitoral winking, and deviating the tail from the perineum. Both first single signs of oestrus and distinct signs were assumed to indicate the first day of the oestrus for the analysis.

The mares were clinically sound and had never had any problems with the reproductive system or behavioural disorders. They had been maintained in the same stable and surrounded by the same horses in neighbouring boxes for over a year. Meadow hay and concentrate were provided to the horses at 6:00, 12:30, and 18:00 daily. The mares were turned out in two constant groups of seven and eight mares into two paddocks after the morning feed for three hours daily. Differences between the social hierarchy status of particular mares within each group were not distinct. There were no strongly dominant, subservient, or solitary mares within either group. The mares were used for leisure riding for 1 h in the afternoon, six days per week, until the observations began.

### 2.2. Experimental Procedure and Measurements

The study was conducted for 5 weeks starting in May. Each mare was examined on 11 days during one oestrous cycle: from the first day of oestrus until the last day before the next oestrus (5 days) and then every third day of dioestrus (6 days) (Table 1). Thus, the examination for the study was not begun at the time of ovulation (usually considered to be day 0 of the cycle) but at the beginning of oestrus. It was more justified to start at the beginning of oestrus because the oestrous and dioestrous variables were the subject of the study and not the ovulation. There were 12 days of difference at the beginning of the oestrus between the first and the last mare. The ultrasonography was conducted in each mare beginning with the third day of oestrus and then on the fourth and possibly fifth day of oestrus, until the ovulation was confirmed [23,24,25]. Seven mares ovulated on the fourth day and eight on the fifth day of oestrus. The day of ovulation was considered to be day 0 of the oestrous cycle. The days preceding the ovulation were numbered −1, −2, −3, and −4, whereas those after ovulation were numbered sequentially. The outcomes from successive days of the examination have been presented in figures.

The cardiac variables in the horses were measured in the box, at rest and 1 h after feeding (7:30). After a 5 min break, the mare was led to the stable corridor that complied with the requirements for thermography tests: an average ambient temperature of 21.2 ± 1.5 °C, 0 m/s air speed, 42.5 ± 5.5% relative air humidity, and a hardened floor [26,27]. The images of superficial temperature were taken simultaneously with the measurement of the rectal temperature. The horse was then returned to the box to perform the test of behaviour towards a human. The mares of the first group were released into the experimental paddock simultaneously, at 8:00, for 60 min. The second group was released after the first one, at 9:00. The two groups of mares had to be studied one by one in the same paddock to limit the number of observers to two. Behavioural observations of the relationship to conspecifics and locomotor activity in mares of the oestrous cycle studied at that time were conducted. Thus, some mares in the groups were turned out only to maintain the stability of the groups for the whole experiment and were not studied on a given day. After the observations in the experimental paddock, both groups of mares were led to grassy paddocks or pastures for 2 h. A similar procedure was used in the evening: feeding, cardiac, and temperature measurements and the test of behaviour towards a human.

The characteristics of the oestrous cycle in mares included cardiac activity variables, body temperature, behaviour in relation to humans and conspecifics, and locomotor activity.

#### 2.2.1. Cardiac Variables

The following cardiac activity variables were analysed: the heart rate (HR; beats per min) in the morning (HRm) and evening (HRe), the root mean square of the successive differences in beat-to-beat intervals (RMSSD; ms) in the morning (RMSSDm) and evening (RMSSDe), and the ratio of spectrum density power from the low-frequency spectrum (LF; ms^2^) to the high-frequency spectrum (HF; ms^2^) of beat-to-beat intervals in the morning (LF/HFm) and evening (LF/HFe). The HR increases as a result of elevated sympathetic nervous system activity, augmented RMSSD shows the predomination of parasympathetic nervous system activity, whereas the LF/HF is an indicator of the functional sympathetic–parasympathetic balance [28]. The 5 min measurements were recorded with Polar RS800CX devices (Polar Elektro Oy, Kempele, Finland). The monitor was attached to the left side of the horse’s body with a rubber band placed around the girth. The records were analysed using PolarProTrainer 5.0 software (Kempele, Finland).

#### 2.2.2. Body Temperature

The rectal temperature (°C) in the morning and evening was measured with a rectal KRUUSE Digi-Vet SC 12 thermometer. The images of superficial temperature (°C) in the morning and evening of the left side of the horse were acquired with a Fluke type Ti 9 infrared thermography camera (Fluke Corporation, Everett, WA, USA). The camera was located on a tripod, 150 cm from the floor. The horses were scanned from a distance of 3 m. The superficial temperature records were analysed using Fluke SmartView 4.3 (Eindhoven, The Netherlands) software. The body surface temperature was the average temperature of the measurements from five regions of interest (ROIs): ROI1—head and neck (the vertical cut-off line at the shoulder joint); ROI2—trunk (vertical cut-off lines at the shoulder joint and the tubercle of the iliac crest, the horizontal cut-off line at the line of the sternum); ROI3—croup (the vertical cut-off line at the tubercle of the iliac crest, the horizontal cut-off line at the stifle joint); ROI4—foreleg (the horizontal cut-off line at the height of sternum); and ROI5—hind leg (the horizontal cut-off line at the stifle joint) [26,29]. Infrared thermography is used as a research and clinical tool both in human and veterinary medicine [29]. The superficial temperature has been used in our study along with the rectal temperature to detect any possible thermal changes that occur during the oestrous cycle. Increases in body temperature correlate with increases in anxiety [30].

#### 2.2.3. Behaviour towards humans

The test of stroke acceptance (active human) showed the behaviour of the mares towards Humans. It was performed in the morning and, for the second time, in the evening, by one experimenter familiar to the mares (A.W., author). She approached at a distance of 0.5 m from the left side of the horse and, after 60 s, tried to stroke the area of the horse’s shoulder. Next, she moved away from the horse (or the horse moved away) to a distance of over 1 m and then approached the horse for a second time to 0.5 m to try stroking again. Two similar trials of stroking the nose and nostrils were conducted immediately after two trials of stroking the shoulder. The scores for the behavioural response of the horses to the test are presented in Table 2.

#### 2.2.4. Behaviour towards Conspecifics, Locomotor Activity and Standing

The mares were released together into a familiar, sandy paddock. The size of the paddock (20 × 25 m) agreed with sizes commonly applied in horse management [31]. The paddock was situated near the stable, away from any roads, noise, or other stressors. Water and food were not available in it. The behaviour of mares was observed by two of the authors (A.W. and B.K.) who sat still outside the fence and were not visible to the horses. One observer recorded behaviours towards conspecifics by counting the number of affiliative (positive) and agonistic (negative) behaviours displayed by mares during the 60 min. Calm approaching and allogrooming were counted among positive behaviours, whereas the following behaviours were considered negative: ears laid back, chasing, bite threat, and kick threat (Table 3) [19,20]. The time of behaviours was also recorded and, in the case of prolonged behaviours, the time was divided by 3 s. Thus, a behaviour, for example, that lasted 6 s was counted as two behaviours. When a horse presented a few behaviours simultaneously, all of them were recorded. Sexual behaviours, such as clitoral winking and deviating the tail from the perineum, were not recorded [6].

Another observer recorded the time of standing (s) of the mares with a SMJ Sport JS−320 stopwatch (Reguły, Poland). Ambulating and single strides (treads) were considered as locomotor activity [32]. The kind of movement was not taken into account. The time of each standing was totalled and the difference between 60 min and the total standing was considered as the locomotor activity. Prolonged locomotor activity was assumed to be a sign of agitation [21].

### 2.3. Statistical Methods

Due to considerable changes in the variables that occurred at the time of ovulation, the measurements obtained during four days preceding the ovulation (from day −3 to 0) and four days of the middle dioestrous period (day 4, 7, 10, and 13) were considered typical for oestrus and dioestrus, respectively. Hence, only these results were taken into account in the statistical analysis. The data were tested regarding the normality of distribution using the Shapiro–Wilk test. The latter did not reject the normality of data in any case. Hence, the analysis of variance for repeated measures (ANOVA GLM) was used for HR, RMSSD, LF/HF, rectal temperature, superficial temperature, and behaviour towards humans regarding the stage of the cycle (oestrus/dioestrus) and time of the day (morning/evening). The Student’s *t*-test for dependent variables was used to analyse the significance of differences between the means of the oestrus and dioestrus for variables measured only once daily: behaviour towards conspecifics and standing and locomotor activity variables. The oestrous/dioestrous and morning/evening dependences between the variables were analysed with Spearman’s rank correlation. Regression models for HR, behaviour towards humans, and standing and locomotor activity were determined to illustrate the dynamics of changes across successive days of the cycle. The significance of the models was tested with the Fisher–Snedecor test, while the degree of fit between the models and the data was analysed using the R2 determination coefficient. Only statistically significant regression models have been presented. The statistical evaluation of the data was performed using Statistica 13.3 and Excel software, assuming the significance level α = 0.05.

## 3. Results

*p*-values of effects on the traits and means of variables are presented in Table 4. According to ANOVA, the effects of the oestrous cycle stage and time of day were statistically significant for HR, RMSSD, rectal temperature, superficial temperature, behaviour towards humans, and negative behaviour towards conspecifics (*p* < 0.05) and not significant for LF/HF and positive behaviour towards conspecifics (*p* > 0.05; Table 4). The interactions between the cycle and time of day were significant for RMSSD, rectal temperature, and superficial temperature (*p* < 0.05).

The HR was higher in oestrus than dioestrus, especially when measured in the evening, and was usually higher in the evening than the morning (*p* < 0.05; Table 4 and Table 5; Figure 1). A drastic drop of the HRe led to a similar value to that of the HRm on day 1 of the oestrous cycle. According to the line regression models determined for the dioestrus, the HRm decreased by 0.07 and HRe increased by 0.08 on every three successive days of dioestrus.

The RMSSD was also higher in oestrus than dioestrus, particularly in the evening (*p* < 0.05; Table 4 and Table 5; Figure 2). The RMSSD was usually higher in the morning than the evening. Both RMSSDm and RMSSDe dropped radically on day 1 of the oestrous cycle. The dioestrus began with a rapid increase in RMSSDm and RMSSDe.

The changes in LF/HF between oestrous and dioestrous and morning and evening measurements were not statistically significant (Table 4 and Table 5). The rectal temperature was significantly higher in oestrus than dioestrus and higher in the evening than the morning (*p* < 0.05; Table 4 and Table 5; Figure 3). The rectal temperature was definitely higher than the superficial temperature. The latter was also significantly augmented in oestrus compared to dioestrus and higher in the evening than the morning (*p* < 0.05; Table 4 and Table 5; Figure 3). The superficial temperature both in the morning and evening dropped rapidly on day 1 of the oestrous cycle.

The mean scores for the behaviour towards humans were increased in oestrus compared to dioestrus and usually higher in the morning than the evening (*p* < 0.05; Table 4 and Table 5; Figure 4). Both morning and evening behaviour towards humans dropped distinctly on the first few days of dioestrus. The line regression models determined for this trait during oestrus showed that the scores increased on average by 0.52 and 0.71, respectively, on each successive day.

The number of positive behaviours towards conspecifics depended on the stage of the oestrous cycle (*p* < 0.05; Table 4 and Table 5; Figure 5). The most frequent positive behaviours were recorded on day 0 of the cycle. The positive behaviours occurred many times more frequently than negative behaviours. The negative behaviours towards conspecifics were less frequent in oestrus than dioestrus (*p* < 0.05; Table 4 and Table 5).

Complementary time traits of standing and locomotor activity differed significantly between oestrus and dioestrus (*p* < 0.05; Table 4 and Table 5; Figure 6). The mares remained standing for a longer time during oestrus than dioestrus, whereas the locomotor activity was lower in oestrus than dioestrus. The standing time lowered dramatically on day 1 of the cycle, whereas the changes in the locomotor activity were the opposite. The line regression models showed that the standing lowered and locomotor activity increased by an average of 0.55 s every three successive days of dioestrus.

Many traits studied correlated significantly with each other (*p* < 0.05; Table 5). More significant correlations were found during oestrus (45) than dioestrus (31). Considering the whole oestrous cycle, most correlations concerned temperatures (sequentially: morning rectal, morning superficial, evening superficial, and evening rectal), then RMSSDm and HRe. The morning rectal temperature, morning and evening superficial temperatures, HRe and RMSSDe during oestrus correlated with most other traits. In turn, the number of negative behaviours towards conspecifics, RMSSDm, and morning and evening rectal temperatures frequently correlated with other traits in dioestrus. There were no significant correlations with LF/HF.

## 4. Discussion

The results of the study show that a lot of the physiological parameters studied were higher in the evening than in the morning (HR, rectal temperature, and superficial temperature) and higher during oestrus than dioestrus. Only RMSSD, the parameter reflecting the parasympathetic nervous system activity, was lower in the evening compared to the morning measurements, however, similar to the other parameters mentioned, it was higher during oestrus than dioestrus. Physiological processes are generally affected by a variety of temporal programmes, such as circadian rhythms, oestrous cycles, and circannual rhythms [33]. Circadian rhythms occur in many behavioural and physiological processes, including the sleep-activity cycle, hormone secretion, metabolism, and body temperature [34]. The rhythmicity associated with the oestrous cycle has been investigated less. In the present study, the changes of physiological variables during the oestrous cycle in mares were significant. Interestingly, changes in the behavioural parameters were also dependent on the oestrous cycle. During oestrus, mares showed a better behaviour towards humans and other horses from the herd and were less locomotively active than during dioestrus. A drastic drop of values in most of the parameters studied was found on day 1 of the oestrous cycle, whereas the score for behaviour towards humans dropped and the number of negative behaviours towards conspecifics increased after oestrus (shown on day 4 of the cycle). The significant regressions and correlations between some of the parameters also confirm the complex mechanism of changes that occur in the mare during the oestrous cycle.

The oestrous period is characterised by high levels of oestrogen in the blood and characteristic sexual behaviour, while dioestrus is accompanied by a high progesterone level. Steroid hormones generally affect cells through a genomic mechanism that regulates gene expression. However, oestrogen can also act through non-genomic mechanisms using membrane oestrogen receptors, which affect the synaptic activity of neurons and, thus, directly influence behaviour. Due to such a wide range of acting on cells, oestrogen increases the synthesis of some neuromediators, such as oxytocin and dopamine in the brain, and stimulates the synthesis of sex hormone binding globulin, which plays a role as a signalling substance in the brain [35]. Oestrogens also regulate the activity of endocrine glands that decrease aldosterone production and increase adrenomedullary catecholamine synthesis [36]. In addition, oestrogens inhibit the release of thyroid hormones, whereas progesterone acts as a stimulant. Changes in the blood thyroid hormones level influence the metabolism and general activity of the body. All these mechanisms are involved in the regulation of sexual behaviour in females and may affect the overall rhythmicity of the oestrous cycle.

As was expected, HRe is higher than HRm, which can be due to a circadian rhythm connected with the daily activity and stress, on the one hand, and nightly tranquillity, on the other. The same factor may affect the degree of oestrous rhythmicity, which is less distinct in HRm than HRe. Both HRm and HRe increase in response to oestrus, which reflects an increased level of arousal before and at the moment of ovulation. The most interesting fact is that HRe decreases sharply at the end of oestrus. It seems that such an HRe drop illustrates a decrease in sympathetic nervous activity 1–2 days after ovulation. In dioestrus, HRe normally rises again, whereas HRm usually decreases before the next oestrus. Evans et al. [24] did not find any consistent effect of ovulation on the HR. However, that study was performed under different conditions and the HR in four mares examined was higher than in our investigation. The influence of the circadian rhythm on the HR in horses was partially confirmed in the study by Janczarek et al. [37], in which higher values of HR were found in old horses during the night period than the light of day.

The HR should be discussed simultaneously with other HR variability parameters since they altogether show changes in the sympatho–vagal balance of the activity of the nervous system. An elevated HR illustrates a shift towards the sympathetic nervous system, whereas increased RMSSD indicates a shift towards the parasympathetic nervous system [38,39]. In the present investigation, the RMSSDm and RMSSDe changed significantly regarding both the time of day and stage of cycle. The drops in RMSSDm and RMSSDe accompanied a drop in HRe after ovulation. This may mean that the ovulating mare undergoes a specific overall slowdown of the activity of both nervous systems. The confirmation of this assumption may be the lack of differences in the LF/HF during the oestrous cycle. This parameter was used in another study on horses as a parameter reflecting the adrenergic–vagal balance [37]. It showed a tendency to increase during the night. Nevertheless, in the present study, both sympathetic and parasympathetic nervous systems showed a little higher activity during oestrus than dioestrus, but the sympatho–vagal balance remained unchanged.

The rectal temperature shows small but statistically significant oestrous cycle-dependent and circadian oscillations. This is similar to women and dairy cattle, in which the body temperature changes immediately prior to ovulation [15]. Ammons et al. [40] did not observe any change in the rectal temperature that could be utilised to predict oestrus or ovulation in mares. Bowman et al. [15] did not find any difference in the rectal temperature dependent on the presence or absence of a follicle either. However, the temperatures measured with an implanted microchip were higher prior to ovulation compared to the time immediately following ovulation. In our study, the oestrous cycle-dependent differences in the rectal temperature were overlapped by the circadian rhythm-associated fluctuations. Thus, our findings do not indicate that the rectal temperature might be a useful tool to detect ovulation in mares.

Studying the mare’s circadian rhythm, Piccione et al. [41] found that the rectal temperature in mares increases during the day and the range of excursion amounts to 1 °C. Although the endogenously generated circadian rhythm is observed, horses, as endothermic homeotherms, can maintain a relatively stable body temperature because of their ability to modulate metabolic heat production as well as convective and evaporative heat-loss [42].

It is worth emphasising that the oestrous cyclicity of the superficial temperature is well-pronounced. Thus, the thermographic technique turns out to be an accurate tool for measuring the temperature changes in the future, although it requires a more advanced appliance and comparable microclimate conditions in which pictures are taken [17,29].

The test of stroke acceptance showed that mares accept being stroked more often in oestrus, particularly in the second part of oestrus, than in dioestrus, and the scores for behaviour towards humans drop in the first days of dioestrus. The mares were habituated to stroking during the whole their lives, hence, the test was not novel for them and the response on subsequent days could not be significantly affected by habituation. It is known that stroking relaxes and rewards an agitated horse or one tired after a stressful situation or physical effort [38]. The changes in the behaviour towards humans during the oestrous cycle may illustrate that mares in oestrus need an assuagement that can be provided by a mild contact with a human. The high scores for the behaviour towards humans in the second half of oestrus also seem compatible with the sexual behaviour, which manifests in searching for contact with a stallion.

The mares studied were accustomed to each other being turned out into paddocks or pasture in the constant social group with an established hierarchy, hence, negative behaviours towards conspecifics occurred rarely. Interestingly, a lower number of negative behaviours in oestrus than dioestrus can also indicate a need for contact with conspecifics during oestrus associated with sexual behaviour. This postulation cannot be confirmed by the number of positive behaviours, which did not show any significant changes during the oestrous cycle.

The longer time of standing in oestrus than in dioestrus shows that, in spite of agitation, mares often remain standing at this time. Such a position is closely associated with sexual behaviour. Standing still with hind limbs spread apart is one of the clearest behaviours that indicate oestrus in the mare [5]. In turn, horses ambulate to a low extent [43]. Only single strides are frequent, which is connected with changing the position of the body and grazing. Our results on a regularly longer time of locomotive activity during dioestrus illustrate that mares become more active in this period. The rapid increase in the time of locomotor activity and decrease in time of standing after ovulation suggest that the prolonged standing in oestrus is due to sexual behaviour.

It is currently known that many of the variables studied correlate with each other in horses. Surprisingly, our results show that the HRm correlates only with the HRe. This can probably be explained by similar HRm values during the oestrous cycle, although the lowering regression in dioestrus is significant. In turn, the HRe correlates with most variables in oestrus, including all temperature variables. Hall et al. [44] reported that the HR correlates with the body temperature and emotional arousal, which, in turn, shows a predominance of sympathetic nervous system activity and manifests with behavioural signs of anxiety. On the other hand, the horse HR and behaviour regarding the interaction with humans depend on horses’ previous experience and training [45,46]. Therefore, some horses can react to stressful stimuli by an elevated HR despite looking calm [47]. Thus, the horse HR and behaviour can change independently of each other. In addition to the HRe, the RMSSDm seems to be an appropriate indicator of the cyclic changes because it correlates with many other traits both in oestrus and dioestrus. In addition, both temperature variables, especially the superficial temperature, are correlated with most other traits. This is consistent with the above-mentioned and other findings on the correlations of horses’ temperatures with arousal [17,30,44,48].

As could have been expected, behavioural traits are more rarely correlated with each other and with physiological variables because of, for example, the character of the measurement. It has been reported that allogrooming reduces emotions in horses, which could be of importance during oestrus [49,50]. In our study, indeed, the allogrooming counted among the affiliative behaviours towards conspecifics, positively correlates with RMSSDm and negatively with morning rectal and superficial temperatures. However, in dioestrus, the outcomes are contrary and difficult to interpret. In turn, negative correlations between affiliative and agonistic behaviours towards conspecifics as well as between the times of standing and locomotor activity are evident because these traits are contrary.

The findings of the study may be important for horse behaviourists and breeders who manage mares in various kinds of use and conduct mating. The outcomes include typical physiological and behavioural means of variables that characterise the process of oestrus and dioestrus. The limitation of the study is the fact that mares vary considerably in the duration of showing signs of oestrus and the moment of ovulation [23]. Hence, in order to ensure the analysis is possible, we had to find mares that were most similar regarding the oestrous cycle process. The mares in the study were examined using ultrasonography only once a day during oestrus, thus, the time of ovulation could not be determined precisely. Therefore, the choice of from day −3 to 0 as representative of oestrus and from day 4 to 13 as representative of dioestrus turned out to be helpful. Such an approach enabled us to omit the transitional period between oestrus and dioestrus when rapid changes of the parameters studied occurred.

## 5. Conclusions

Summing up, the present study shows that the oscillation of physiological and behavioural variables accompanies changes in mares’ sexual behaviour. In oestrus, augmented physiological variables, such as the HR, rectal temperature, and superficial temperature, indicate the elevated activity of the adrenergic nervous system, most often connected with emotional excitability in healthy horses. Opposite to this, levels of RMSSD, behaviour towards humans, behaviours towards conspecifics, and time of standing are not characteristic of an arousal but relate closely to a relaxed state. Thus, the typical sexual behaviour of mares, i.e., staying in place and increased tolerance to stimuli, occurs with the simultaneous stimulation of both sympathetic and parasympathetic nervous systems. Changes in behavioural variables usually follow changes in physiological variables and occur at the beginning of dioestrus. The end of oestrus is manifested by a rapid decrease in the HRe, RMSSDm, and RMSSDe, as well as morning and evening superficial temperatures, whereas, at the beginning of dioestrus, the scores for morning and evening behaviour towards humans are lower and the number of negative behaviours towards conspecifics increases. The time of locomotor activity arises earlier, already at the end of oestrus. The outcomes may contribute to the knowledge of, among others, mare owners and equine behaviourists who evaluate the oestrus solely by a mare’s sexual behaviours without taking into account other rhythmically changing variables.

## Figures and Tables

**Figure 1 animals-13-00211-f001:**
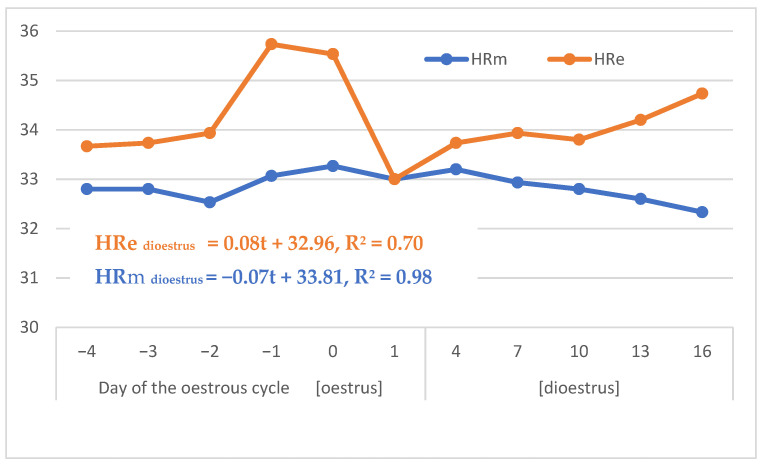
Dynamics of HR changes in the morning and evening during the oestrous cycle and regression models for dioestrus. HR—heart rate (beats per min).

**Figure 2 animals-13-00211-f002:**
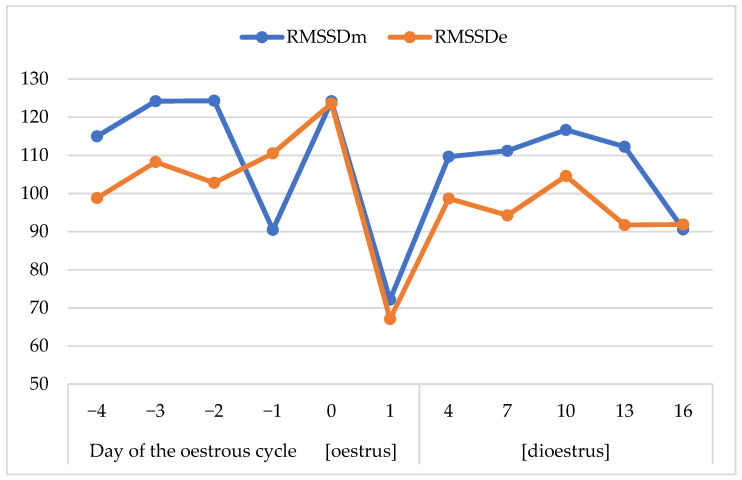
Dynamics of RMSSD changes in the morning and evening during the oestrous cycle. RMSSD—root mean square of successive differences in beat-to-beat intervals (ms).

**Figure 3 animals-13-00211-f003:**
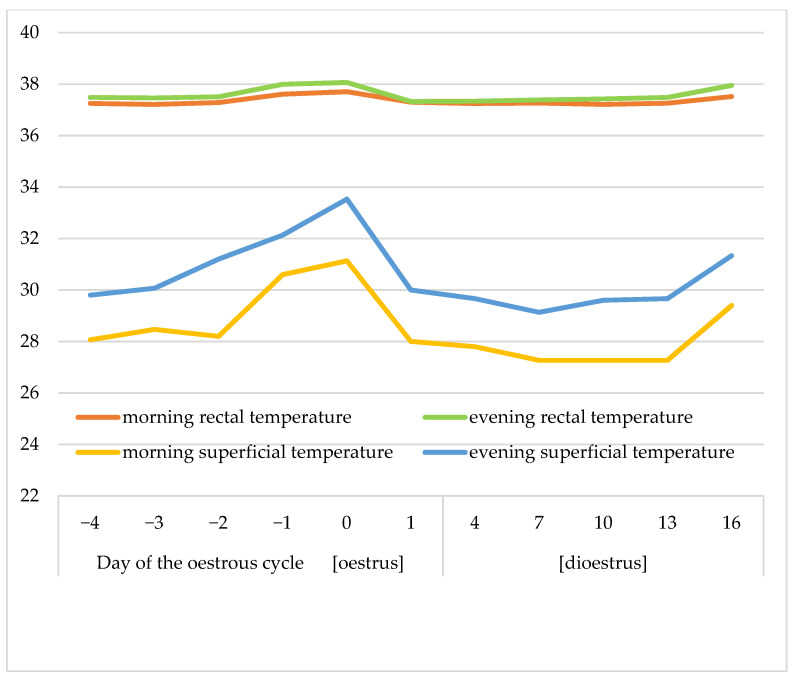
Dynamics of changes in mean rectal and superficial temperatures (°C) in the morning and evening during the oestrous cycle.

**Figure 4 animals-13-00211-f004:**
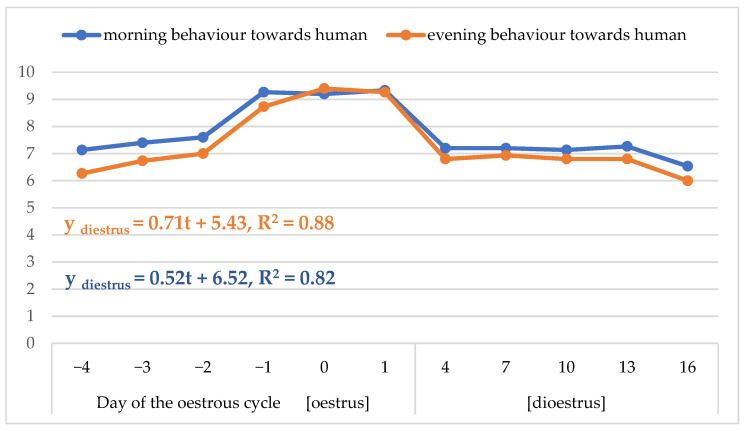
Dynamics of changes in mean scores for the behaviour towards human in the morning and evening during the oestrous cycle and regression models for oestrus.

**Figure 5 animals-13-00211-f005:**
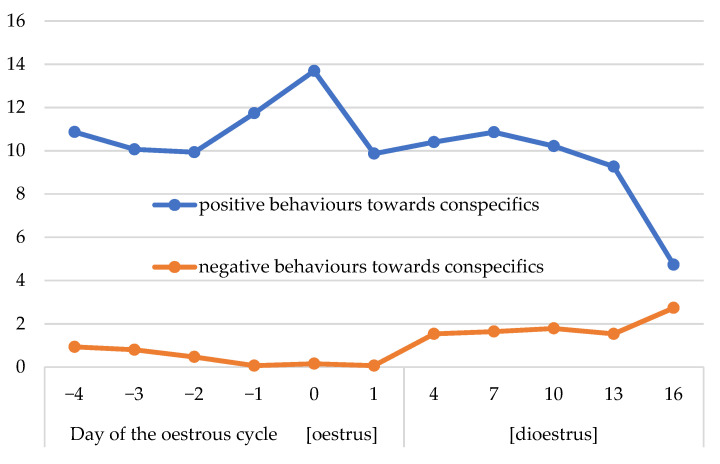
Dynamics of changes in mean scores for the frequency of positive and negative behaviours towards conspecifics in the morning during the oestrous cycle.

**Figure 6 animals-13-00211-f006:**
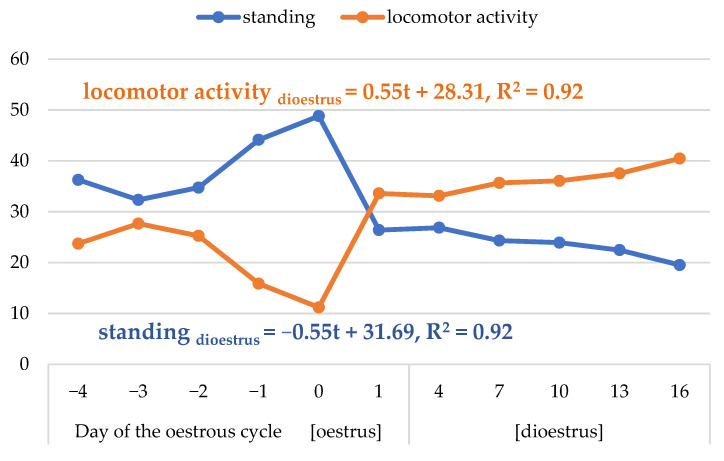
Dynamics of changes in the average time of standing and locomotor activity (s) in the morning during the oestrous cycle and regression models for dioestrus.

**Table 1 animals-13-00211-t001:** The schedule of examining the mares.

Day of Examination	Day of Oestrous Cycle	Presence of Heat Signs	Results Extracted for Statistical Analysis
1	−4	+	−
2	−3	+	Oestrus
3	−2	+	
4	−1	+	
5	0	+	
6	1	+	−
7	4	−	Dioestrus
8	7	−	
9	10	−	
10	13	−	
11	16	−	−

**Table 2 animals-13-00211-t002:** Score scale for the response of mares in the test of stroke acceptance.

Scores for Behaviour	Shoulder		Nostrils	
	First Trial	Second Trial	First Trial	Second Trial
1	−	−	−	−
2	−	+	−	−
3	+	−	−	−
4	+	+	−	−
5	−	+	−	+
6	−	+	+	−
7	+	+	−	+
8	+	+	+	−
9	−	+	+	+
10	+	+	+	+

“+”—a mare allowed the stroking, “−“—a mare did not allow the stroking.

**Table 3 animals-13-00211-t003:** Description of the behaviours towards conspecifics, recorded in the study ([19,20]; modified).

Behaviour	Description
	**Positive behaviours**
Calm approaching	Calm approaching another mare to remain in proximity.
Allogrooming	Two mares standing beside one another, grooming each other’s chosen part of the body by gentle nipping, nuzzling, or rubbing.
	**Negative behaviours**
Ears laid back	The head turned towards another mare with ears pressed caudally against the head and neck.
Chasing	Running after another mare, often trying to bite or strike the other mare, while the latter is running away.
Bite threat	Rapid turning of the head towards another mare and opening mouth.
Kick threat	Rapid extension of one or both hind limbs, trying to hit another mare.

**Table 4 animals-13-00211-t004:** *p*-values of effects (stage of the oestrous cycle, time of day, and interaction) on the traits and means of variables (means ± SD).

Variables	*p*—Stage of Cycle	*p*—Time of Day	*p*—Stage of Cycle × Time of Day Interaction	Morning		Evening	
				Oestrus	Dioestrus	Oestrus	Dioestrus
HR	<0.0001 *	<0.0001 *	0.6333	32.9 ± 0.88 ^A^	32.88 ± 1.23 ^A^	34.7 ± 1.14 ^B^	33.92 ± 1.02 ^C^
RMSSD	<0.0001 *	<0.0001 *	0.0024 *	115.8 ± 18.60 ^A^	112.5 ± 8.85 ^A^	111.3 ± 13.60 ^A^	97.3 ± 12.00 ^B^
LF/HF	0.567	0.368	0.871	186 ± 91.00	219 ± 86.80	212 ± 103.00	254 ± 109.00
Rectal temperature	<0.0001 *	<0.0001 *	<0.01 *	37.46 ± 0.23 ^A^	37.25 ± 0.07 ^B^	37.76 ± 0.29 ^C^	37.41 ± 0.10 ^A^
Superficial temperature	<0.0001 *	<0.0001 *	<0.041 *	29.60 ± 1.50 ^A^	27.40 ± 0.61 ^B^	31.73 ± 1.53 ^C^	29.52 ± 0.88 ^A^
Behaviour towards humans	<0.0001 *	<0.0001 *	0.5032	8.37 ± 1.21 ^A^	7.20 ± 0.79 ^Bb^	8.80 ± 6.14 ^A^	6.83 ± 0.69 ^b^
Positive behaviours towards conspecifics	0.1531			11.73 ± 5.82	10.50 ± 4.16		
Negative behaviours towards conspecifics	<0.0001 *			0.37 ± 0.87 ^x^	1.60 ± 1.16 ^y^		
Standing	<0.0001 *			40.80 ± 9.54 ^x^	24.18 ± 4.93 ^y^		
Locomotor activity	<0.0001 *			20.02 ± 9.53 ^x^	35.82 ± 4.93 ^y^		

*p*—probability value; HR—heart rate (beats per min); RMSSD—root mean square of successive differences in beat-to-beat intervals (ms); LF/HF—ratio of spectrum density power from low-frequency (LF; ms^2^) to high-frequency (HF; ms^2^) of beat-to-beat intervals; * statistically significant effect (*p* < 0.05); ^A, B, C^—means in rows marked with different capital letters differ significantly according to Tukey’s test at *p* < 0.001, ^b^—marked with small letter differ at *p* < 0.05; ^x, y^—means in rows marked with different letters differ significantly according to Student’s *t*-test (*p* < 0.001).

**Table 5 animals-13-00211-t005:** Significant correlations (*p* < 0.05) between traits: bottom—during oestrus period, top—during dioestrus period.

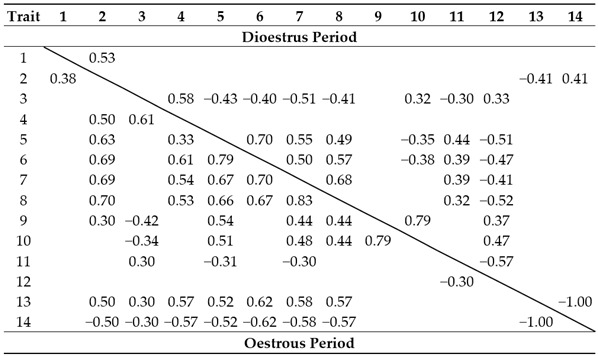

1—HRm, 2—HRe, 3—RMSSDm, 4- RMSSDe, 5—morning rectal temperature, 6—evening rectal temperature, 7—morning superficial temperature, 8—evening superficial temperature, 9—morning behaviour towards humans, 10—evening behaviour towards humans, 11—positive behaviours towards conspecifics, 12—negative behaviours towards conspecifics, 13—standing, and 14—locomotor activity.

## Data Availability

The data presented in this study are available on request from the corresponding author.

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
