# Peer review of "Variation of Physiological and Behavioural Parameters during the Oestrous Cycle in Mares"

_animals, 2023, doi:10.3390/ani13020211_

Round 1
Reviewer 1 Report
General Comments:
This is an interesting study and probably of more interest to mare owners and equine behaviorists than breeders and reproductive specialists, as the title suggests. The heart rate data and behavioral tests are appropriate parameters for this study, however there may be other parameters that would add more data to the study. Did the authors consider using pedometers or GPS locators to monitor daily activity during estrus and diestrus for a longer period of time? The graphs are easy to interpret, however the authors should consider different line styles (rather than different colors) for the different groups so that they are more easily distinguished from each other when printed or copied in black and white. Since the study is comparing differences between estrous and diestrous mares the authors really need to add discussion relevant to the major hormones and their interactions. Certainly this is an area that lacks scientific studies but a discussion that at least involves the relationship between the gonadal axis hormones and: thyroid hormones, growth hormone/IGF, and renin-angiotension-aldosterone system would add value and perhaps spark additional research in horses to explore those possible interactions.
Specific Comments:
Line 20 It may be more clear to just list the days of diestrus (ex. D1, D4 etc., if this is what was done)
Line 21 Delete ‘Results show that the…. “ Heart rate, rectal and surface temperatures etc.”
Line 23 Is there a better word choice than ‘attitude’ as it is not described and likely the word limit on the abstract doesn’t allow it.
Line 26 Unclear
Line 27 Is ‘inter alia’ a universally understood Latin phrase??
Line 29 Change oestrus to oestrous (as it is being used as an adjective)
Line 31 Change analyse to analyze
Line 34 Reword for clarity. Did the authors observe the mares on the 1st day of diestrus and then every 3rd day of diestrus for a total of 5?
Line 40 Authors should be consistent with verb tense (present vs. past)
Line 88-92 Reword sentence ex. Changes in the mare’s physiological indices etc…..may provide useful information to improve animal welfare and human safety (??)
Line 102 Delete ‘high’ , as I’m not sure what this means without an explanation
Line 103 Change everyday to daily
Line 105 Delete ‘the following’ (and just list them behind :)
Line 120 Why not just say the study was carried for 5 (or 6 weeks) starting in May (state year)
Line 189 Sentence is unclear. What does the test of an active human mean? Is there a better way to describe the test (ex. Neck stroke acceptance?). Did the authors stroke the nose or the nostrils?
Table 2 Is the first column of 1-10 the ID of each mare? If so, shouldn’t this be 1-15? Or is it ‘day’??
Line 202 Commonly used for what?? Turnout??; Authors should also state stocking density in square meters per animal
Table 4 Can the information from table 4 be combined with table 5? Table 5 does a better job at reporting differences between variables based on both time of day and stage of cycle. This can be seen by looking at table 5 and then understanding why there were no interaction effects for HR and behavior towards humans. (per table 4) But it is cumbersome.
Line 282 Delete ‘morning’
Table 6 Table 6 is somewhat interesting it doesn’t add value to the data. It is not easily read. Suggestion is to eliminate table 6.
Line 352 Delete inter alia
Line 418 Change higher to longer
Discussion The discussion would be strengthened if the authors at least bring in the interaction of reproductive hormones (estrogens, LH) with other hormonal systems. Certainly the literature in horses is lacking, there is data that supports the complex interactions of these systems. Examples include 2 types of estrogen receptors and where they are found; possible interactions with thyroid system and renin-angiotension systems.
Author Response
Thank you very much for all the comments. They helped a lot to improve the manuscript.

Reviewer 2 Report
-The study by Stachurska et al. addressed the changes in physiological and general behavior parameters during an estrous cycle in mares. Although the differences in reproductive behavior between estrus and diestrus in mares are well known, there is paucity of information on the topic chosen by the researchers for this study.
-In general, the study is well designed. However, the sample size is small.
-The manuscript needs editing to make it clear. Some parts of the manuscript are difficult to follow. For instance, “Fifteen adult mares were examined in the morning and evening, on six days of oestrus, then on five days every three successive days of dioestrus.”
-There are also some inconsistencies in the citation of references within the text. For instance, in Line 362, the reference is cited as Janczarek et al. [2020].
-I suggest replacing “ovulatory cycle” with “estrous cycle” throughout for accuracy.
-Tables should be checked again for formatting errors (superscripts etc.) and any abbreviations used should be explained in the footnotes. Abbreviations used in the figures should be defined in the legends. Also, the authors should add the measures of spread for the variables shown in the figures.
-There are several grammatical and typographical errors in the manuscript.
Author Response
Thank you for the review. The manuscript has been corrected according to all comments
